# Theoretical Model and Solution of Dynamic Evolution in Initial Stage of Lacustrine Delta

**Weiyan Xin [1], Yanhua Yang [1], Wude Xie [2], Ziqing Ji [3],\* and Yuchuan Bai [3]**

[1]   Tianjin Research Institute for Water Transport Engineering, Ministry of Transport, Tianjin 300456, China
[2]   Marine College, Shandong University, Weihai 264209, China
[3]   Institute for Sediment, River and Coast Engineering, Tianjin University, Tianjin 300072, China
\*   Correspondence: ziqing_ji@163.com

**Abstract:** Sediment is carried by flows into lake during the formation of lacustrine deltas. The subsequent movement of sediment-laden flow is maintained by the initial momentum at the early stage of delta growth. A theoretical model of the plane jet boundary layer in the initial stage of a lacustrine delta, including the consideration of gravity action on the slope and bottom friction, was established according to characteristics of this process, and a similarity solution method was used to solve the model. A coefficient $\varepsilon$ was introduced that describes the characteristics of sediment-laden flow spreading along a straight line and its value is given. The theoretical expression of morphological characteristics of the initial stage was derived based on the general mathematical model of river bed evolution, and the erosion and deposition patterns were quantitatively analyzed. It was verified through experimental data that the theoretical model can be solved accurately and, can satisfactorily describe the trend in erosion and deposition and morphological characteristics of the initial stage of a delta.

**Keywords:** theoretical model; dynamic evolution; initial stage; lacustrine delta; jet boundary layer





## 1. Introduction

A delta is a kind of deposit formed by the deposition of sediment carried by flow when it enters lake or sea. Movement in the sediment-laden flow can be described as a turbulent jet process, i.e., a diffusion process of unconfined flow that decelerates towards the receiving water under the action of bottom friction and lateral diffusion [1,2]. Deltas have unique geographical advantages and play a crucial role in water conservation, climate regulation, energy supply, etc. Therefore, it is of great significance to explore the process and mechanism of the growth and evolution of deltas.

Bedload is an important form of sediment movement into lakes, and has an important impact on the formation and evolution of deltas. For example, according to the study of Chen et al. [3], the reduction in bedload and bottom sediment from June 2003 to December 2010 accounted for about 7.3% of the suspended sediment in the Three Gorges reservoir. It can be seen that the bedload is an important type of sediment movement in the reservoir area, and its movement will inevitably have an important impact on the formation of the reservoir delta. Their study also shows that the proportion of bedload is affected by sediment particle size, water depth, and flow velocity. Taking the sediment movement in the fluctuating backwater area of the Three Gorges Reservoir as an example, under the normal flow conditions in the fluctuating backwater area of the Three Gorges reservoir (the average water depth is greater than 20 m and the average flow velocity is less than 2.0 m/s), for different sediment particles, the larger the sediment particle size, the larger the corresponding proportion of bedload. Coarse sediment is dominated by bed motion. The smaller the sediment particle size, the smaller the corresponding proportion of bedload. Fine sediment is dominated by suspense motion. With the increase in the water depth in the fluctuating backwater area, the flow velocity decreases, and the percentage of sediment transport increases; that is, part of the suspended load is transformed into bedload. Under



the condition of an average depth of 20 m and an average flow velocity of 2.0 m/s, the percentage of bedload for sediment with a particle size of 1.0 mm is 50%, and is 13% for sediment with a particle size of 0.1 mm, whereas the value is basically 0 for sediment with a particle size of less than 0.03 mm [3]. It should also be noted that it is difficult to observe the bedload, and it is easy to miss the measurement, resulting in the measured bedload sediment transport rate being small in practice. Relatively speaking, it is necessary to study the movement rule of the bedload of deltas from a theoretical point of view.

Jet theory is widely used to describe the process of a river entering a static water body [4–6]. The basic principle of a turbulent jet is analyzed by Abramovich [7] and Rajaratnam [8]. The jet flow dynamics of a delta can be mathematically described by three-dimensional turbulent equations, which are time averaged, non-uniform, and incompressible [9]. The three-dimensional equation can be simplified to a two-dimensional equation that can still describe the physical phenomenon accurately in some specific cases. The turbulent jet in shallow water can be described by Navier–Stokes equations, which are time- and depth-averaged [9]. Velocity self-similarity along the longitudinal section is the basis of the classical turbulent jet [7,9] and experimental results showed that the similarity function is valid in the case of a free jet [10]. Nardin et al. [5] proposed that the flow showed the characteristics of combining free turbulence and wall shear turbulence when the jet boundary was limited. In addition, the turbulent jet can be divided into two parts: one close to the entrance where axial velocity can be assumed to be constant and the kinetic energy of the flow dissipates rapidly, and the other, which is far from the entrance where similar characteristics of velocity in the cross-section appear. Özsoy [11] proposed that the velocity distribution and the half width of the jet satisfy a certain function.

The exchange characteristics at tidal inlets were analyzed by Taylor and Dean [12] by assuming no entrainment and constant water depth. Wang [13] applied the theory of a plane turbulent jet to explain the phenomena of water and sediment in an estuary, combining the continuous equation and momentum equation with the sediment diffusion equation, and the analytical solutions of velocity and sediment concentration in a section were obtained by similarity functions.

Through theoretical derivation and field data verification, Muto and Steel [14] concluded that the coastline of a delta would inevitably retreat when the relative water surface downstream continued to rise at a constant rate and the movement rate of the delta front changed with time while other conditions remained unchanged. Parker and Muto [15] calculated the change in the delta coastline during sea level rise through a one-dimensional mathematical model. Edmonds et al. [16] established a geometric model to calculate the thickness of the topset and foreset of a delta dominated by the topset under different alluvial and water-receiving conditions, so as to determine the environmental conditions for foreset formation to predict the type of delta. Price [17] proposed a model based on random walk theory that can simulate the deposition process of an alluvial fan. Han [18,19] derived the theoretical expression of delta morphology characteristics, such as the aggradation rate of a reservoir delta deposited by suspended sediment, through an unbalanced sediment transport equation. Jimenez-Robles et al. [20] addressed the role of the receiving basin slope in the hydrodynamics of an exiting sediment-laden turbulent jet and the resulting mouth-bar morphodynamics.

At present, few theoretical models have been established and solved according to the characteristics of the initial stage of a delta, and few studies have concentrated on the role of bedload during the formation process of the initial stage. Theoretically exploring this process will help further understand the evolution mechanism of deltas.

## 2. The Theoretical Mode of Plane Jet Boundary Layer in the Initial Stage of Delta

### 2.1. Theoretical Model

Sediment is carried to the receiving water during delta growth and the dynamic decline leads to sediment settlement and deposition. This process is equivalent to the jet movement of one fluid from a small cross-section into another. The velocity of the jet is

high in the initial stage after the flow jetting downstream from the orifice, which mainly depends on the initial momentum to maintain its motion. The initial momentum plays a dominant role in the movement in the initial stage, and the jet continues to move and expand in the subsequent stage. Herein, a theoretical model is established and solved according to characteristics of the movement of flow and sediment in the initial stage.

It is assumed that the initial incident velocity is $u_0$, and the velocity of entrainment and the mixing part between the jet core and the static liquid is less than $u_0$, which is called the jet boundary layer. The jet zone with a longitudinal velocity not equal to zero is a combination of the upper and lower boundary layers bounded by the center line, as shown in Figure 1.

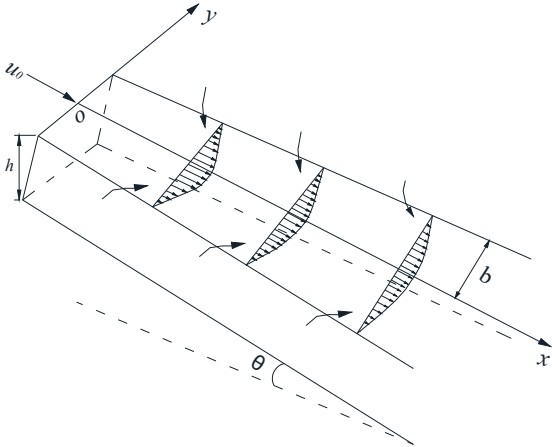

**Figure 1.** The diagram of the plane jet boundary layer in the initial stage of the delta.

The water depth is small relative to the regional scale of the water surface under shallow-water conditions, that is, the vertical scale is far smaller than the horizontal scale. The depth-integrating shallow-water equation of sediment-laden flow is used to describe the movement of flow according to characteristics of shallow water:

$$\frac{\partial}{\partial t}(h\rho_m) + \frac{\partial}{\partial x}(hu_x\rho_m) + \frac{\partial}{\partial y}(hu_y\rho_m) - \frac{\partial}{\partial x}\left(h\varepsilon_x\frac{\partial S}{\partial x}\right) - \frac{\partial}{\partial y}\left(h\varepsilon_y\frac{\partial S}{\partial y}\right) + \rho_0\frac{\partial h_0}{\partial t} = 0 \quad (1)$$

$$\frac{\partial}{\partial t}(hu_x\rho_m) + \frac{\partial}{\partial x}(hu_x^2\rho_m) + \frac{\partial}{\partial y}(hu_xu_y\rho_m) - \rho_mf_xh$$
$$= -\rho_mgh\left(\frac{\partial h}{\partial x} + \frac{\partial y_0}{\partial x}\right) - g\frac{\Delta\rho}{\rho_s}\frac{h^2}{2}\frac{\partial S}{\partial x} + \rho_mhv\left(\frac{\partial^2 u_x}{\partial x^2} + \frac{\partial^2 u_x}{\partial y^2}\right) + \tau_{sx} - \rho_mgh\frac{n^2\sqrt{u_x^2+u_y^2}u_x}{h^{\frac{4}{3}}} \quad (2)$$

$$\frac{\partial}{\partial t}(hu_y\rho_m) + \frac{\partial}{\partial x}(hu_xu_y\rho_m) + \frac{\partial}{\partial y}(hu_y^2\rho_m) - \rho_mf_yh$$
$$= -\rho_mgh\left(\frac{\partial h}{\partial y} + \frac{\partial y_0}{\partial y}\right) - g\frac{\Delta\rho}{\rho_s}\frac{h^2}{2}\frac{\partial S}{\partial y} + \rho_mhv\left(\frac{\partial^2 u_y}{\partial x^2} + \frac{\partial^2 u_y}{\partial y^2}\right) + \tau_{sy} - \rho_mgh\frac{n^2\sqrt{u_x^2+u_y^2}u_y}{h^{\frac{4}{3}}} \quad (3)$$

where $u_x$, $u_y$ are the velocities in $x$ and $y$ directions respectively; $h$ is water depth; $\rho_m, \rho_s, \Delta\rho, \rho_0$ are sediment-laden flow density, sediment density, density difference between sediment and clear water, and mixed density of bed sediment and saturated pore water respectively; $y_0$ is the height of the bed sediment bottom; $h_0$ is the height of the deposit; $S$ is the average sediment concentration of the cross-section; $\varepsilon_x$ and $\varepsilon_y$ are turbulent diffusion coefficients in $x$ and y directions, respectively; $f_x$, $f_y$ are unit mass forces in $x$ and $y$ directions, respectively; $v$ is the dynamic viscosity coefficient; $\tau_{sx}$, $\tau_{sy}$ are wind stresses in $x$ and $y$ directions, respectively; $n$ is the roughness coefficient.

It should be noted that the third-order correlation term is usually not considered when the equation is time-averaged; the terms of convective motion and turbulence pulsation are not considered in the volume of erosion and deposition, but its influence is only considered near the interface. In addition, the time-average values of the terms of pulsation $\overline{\rho'_m u'_x}$ and $\overline{\rho'_m u'_y}$ can be regarded as the diffusion caused by turbulence pulsation in $x$-axis and $y$-axis directions, respectively, which tends to make the uneven muddy water density uniform

along the flow direction. Therefore, they can be related to time-average parameters, yielding $\overline{\rho'_m u'_x} = -\varepsilon_x \frac{\partial \overline{\rho_m}}{\partial x}$ and $\overline{\rho'_m u'_y} = -\varepsilon_y \frac{\partial \overline{\rho_m}}{\partial y}$ [21].

The assumptions of the above formula are as follows: the variation rate of velocity and water depth to time is ignored under the condition of steady flow; derivative terms of viscous stress and pressure change are ignored under the shallow-water condition; wind stress and the Coriolis force are ignored when viscous stress on the bed surface is considered; and sediment-laden flow is considered incompressible, so $\rho_m$ is constant. Thus, the continuity and momentum equations can be simplified as follows:

$$\frac{\partial}{\partial x}(hu_x) + \frac{\partial}{\partial y}(hu_y) = 0 \tag{4}$$

$$\frac{\partial}{\partial x}\left(hu_x^2\right) + \frac{\partial}{\partial y}(hu_x u_y) - hf_x = vh\left(\frac{\partial^2 u_x}{\partial x^2} + \frac{\partial^2 u_x}{\partial y^2}\right) - gh\frac{n^2\sqrt{u_x^2 + u_y^2}u_x}{h^{\frac{4}{3}}} \tag{5}$$

$$\frac{\partial}{\partial x}(hu_x u_y) + \frac{\partial}{\partial y}\left(hu_y^2\right) - hf_y = vh\left(\frac{\partial^2 u_y}{\partial x^2} + \frac{\partial^2 u_y}{\partial y^2}\right) - gh\frac{n^2\sqrt{u_x^2 + u_y^2}u_y}{h^{\frac{4}{3}}} \tag{6}$$

The water depth $h$ is a one-dimensional function of $x$ [13], and the derivative term of water depth $h$ to $x$ can be ignored when the bed slope is small in the initial stage when erosion and deposition are just beginning, so $h$ can be regarded as a constant. Equations (4)–(6) can be further simplified:

$$\frac{\partial u_x}{\partial x} + \frac{\partial u_y}{\partial y} = 0 \tag{7}$$

$$u_x\frac{\partial u_x}{\partial x} + u_y\frac{\partial u_x}{\partial y} - f_x = v\left(\frac{\partial^2 u_x}{\partial x^2} + \frac{\partial^2 u_x}{\partial y^2}\right) - g\frac{n^2\sqrt{u_x^2 + u_y^2}u_x}{h^{\frac{4}{3}}} \tag{8}$$

$$u_x\frac{\partial u_y}{\partial x} + u_y\frac{\partial u_y}{\partial y} - f_y = v\left(\frac{\partial^2 u_y}{\partial x^2} + \frac{\partial^2 u_y}{\partial y^2}\right) - g\frac{n^2\sqrt{u_x^2 + u_y^2}u_y}{h^{\frac{4}{3}}} \tag{9}$$

The geometric scale and flow scale in the plane are $L$ and $U_\infty$ (incoming flow velocity at infinity), respectively, while the vertical geometric scale is $\varepsilon L(\varepsilon \ll 1)$. The above formulas are nondimensionalized:

$$\frac{\partial u_x^*}{\partial x^*} + \frac{\partial u_y^*}{\partial y^*} = 0 \tag{10}$$

$$u_x^*\frac{\partial u_x^*}{\partial x^*} + u_y^*\frac{\partial u_x^*}{\partial y^*} - \frac{f_x^* L^2}{T^2 U_\infty^2} = \frac{1}{Re}\left(\frac{\partial^2 u_x^*}{\partial x^{*2}} + \frac{\partial^2 u_x^*}{\partial y^{*2}}\right) - \frac{g^* L^2}{T^2}\frac{n^2\sqrt{u_x^{*2} + u_y^{*2}}u_x^*}{(h^*\varepsilon L)^{\frac{4}{3}}} \tag{11}$$

$$u_x^*\frac{\partial u_y^*}{\partial x^*} + u_y^*\frac{\partial u_y^*}{\partial y^*} - \frac{f_y^* L^2}{T^2 U_\infty^2} = \frac{1}{Re}\left(\frac{\partial^2 u_y^*}{\partial x^{*2}} + \frac{\partial^2 u_y^*}{\partial y^{*2}}\right) - \frac{g^* L^2}{T^2}\frac{n^2\sqrt{u_x^{*2} + u_y^{*2}}u_y^*}{(h^*\varepsilon L)^{\frac{4}{3}}} \tag{12}$$

The above formula can be approximated when $Re \gg 1$ in the actual situation:

$$\frac{\partial u_x^*}{\partial x^*} + \frac{\partial u_y^*}{\partial y^*} = 0 \tag{13}$$

$$u_x^*\frac{\partial u_x^*}{\partial x^*} + u_y^*\frac{\partial u_x^*}{\partial y^*} - \frac{f_x^* L^2}{T^2 U_\infty^2} + \frac{g^* L^2}{T^2}\frac{n^2\sqrt{u_x^{*2} + u_y^{*2}}u_x^*}{(h^*\varepsilon L)^{\frac{4}{3}}} = 0 \tag{14}$$

$$u_x^*\frac{\partial u_y^*}{\partial x^*} + u_y^*\frac{\partial u_y^*}{\partial y^*} - \frac{f_y^* L^2}{T^2 U_\infty^2} + \frac{g^* L^2}{T^2}\frac{n^2\sqrt{u_x^{*2} + u_y^{*2}}u_y^*}{(h^*\varepsilon L)^{\frac{4}{3}}} = 0 \tag{15}$$

The unit mass force in the $x$-direction is $g \sin \theta$; then, the above formula is written in its dimensional form:

$$\frac{\partial u_x}{\partial x} + \frac{\partial u_y}{\partial y} = 0 \tag{16}$$

$$u_x \frac{\partial u_x}{\partial x} + u_y \frac{\partial u_x}{\partial y} - g \sin \theta + g \frac{n^2 \sqrt{u_x^2 + u_y^2}\, u_x}{h^{\frac{4}{3}}} = 0 \tag{17}$$

$$u_x \frac{\partial u_y}{\partial x} + u_y \frac{\partial u_y}{\partial y} + g \frac{n^2 \sqrt{u_x^2 + u_y^2}\, u_y}{h^{\frac{4}{3}}} = 0 \tag{18}$$

*2.2. Similarity Solution Method of the Model*

Self-similarity is a phenomenon that develops with time. The distributions of system properties at a specified moment and position can be obtained through similarity transformation [22]. The calculation and description of the phenomenon's properties can be simplified by self-similarity, and a group of random empirical data points can be transformed onto a curve or a surface through the selected self-similarity variables. Partial differential equations can be simplified into ordinary differential equations by the similarity solution method.

Similarity solutions exist in some fluid problems with specific physical symmetry. The experimental data also show that the cross-sectional velocity distribution of a plane jet is similar [23–26]. In addition, there should be a similar solution for velocity in the initial stage of the delta formation process.

The experimental observation shows that the change in jet thickness is linearly extended, which has statistical average significance.

As shown in Figure 1, there is a corresponding velocity distribution at each $x$ whose common feature is that the velocity at the outer edge of the jet boundary layer is zero while the velocity at the axis is the largest. The velocity distribution on each cross-section of the jet is similar, which provides a basis for classical turbulent jets [7,9]:

$$\frac{u_x}{u_m} = f\left(\frac{y}{b}\right) \tag{19}$$

where $b$ is the characteristic half-thickness of the jet section and takes the value of $y$, where velocity is equal to $u_x = \frac{u_m}{e}$. At present, some studies [27,28] have given $b = \varepsilon' x$ under the condition of a clear water jet and $\varepsilon' = 0.154$ was obtained based on the experimental results.

We use its definition for reference and introduce a similar coefficient $\varepsilon$ to describe the characteristics of sediment-laden flow spreading along a straight line, and its value was obtained through trial calculations based on our experimental data.

Some studies have chosen the exponential form as the functional form of the velocity profile [29], while some use the polynomial form [30]. The form of the Gaussian normal distribution according to experimental data and the random nature of turbulence is also widely used [28] and was selected in this study, as:

$$u_x = u_m \exp\left(-\frac{y^2}{b^2}\right) \tag{20}$$

We suppose that $\eta = -\frac{y^2}{b^2}$ and $\varphi(\eta) = \exp(\eta)$, and Equation (11) can be expressed as:

$$u_x = u_m \varphi(\eta) \tag{21}$$

Stream function $\psi$ is introduced, and $u_x = \frac{\partial \psi}{\partial y}$ is obtained according to the definition of the stream function. Equation (21) can be substituted into $u_x = \frac{\partial \psi}{\partial y}$ to give:

$$\psi = \int u_x dy = -\frac{\varepsilon^2 x^2 u_m}{2y} \int \varphi(\eta) d\eta \tag{22}$$

We suppose $f(\eta) = \int \varphi(\eta) d\eta$, and according to the definition of stream function, we have:

$$u_y = -\frac{\partial \psi}{\partial x} = \frac{\varepsilon^2 e^\eta}{2y} \left[ u_m \left( 2x + x^2 \right) + x^2 u_m' \right] \tag{23}$$

The momentum equation at the jet axis can be simplified as follows according to the symmetry of the jet:

$$\frac{du_m}{dx} = \frac{g \sin \theta}{u_m} - \frac{g n^2 u_m}{h^{\frac{4}{3}}} \tag{24}$$

Equation (24) is taken into Equation (23) and we have:

$$u_y = -\frac{\partial \psi}{\partial x} = \frac{\varepsilon^2 e^\eta x}{2y} \left[ u_m(2 + x) + gx \left( \frac{\sin \theta}{u_m} - \frac{n^2 u_m}{h^{\frac{4}{3}}} \right) \right] \tag{25}$$

The following can be obtained by integrating on both sides of Equation (24):

$$\ln \left( g \sin \theta - \frac{g n^2 u_m^2}{h^{\frac{4}{3}}} \right) = -\frac{2 g n^2}{h^{\frac{4}{3}}} x + \ln C \tag{26}$$

That is:

$$u_m = \frac{h^{\frac{2}{3}}}{n} \sqrt{\frac{1}{g} C e^{-\frac{2 g n^2}{h^{\frac{4}{3}}} x} + \sin \theta} \tag{27}$$

where $C$ is a constant.

The velocity at the origin is $u_0$, and we have $C = g \left( \frac{u_0^2 n^2}{h^{\frac{4}{3}}} - \sin \theta \right)$, which can be taken into Equation (27); thus, the expression of axial velocity variation along the $x$-axis is obtained:

$$u_m = \frac{h^{\frac{2}{3}}}{n} \sqrt{\left( \frac{u_0^2 n^2}{h^{\frac{4}{3}}} - \sin \theta \right) e^{-\frac{2 g n^2}{h^{\frac{4}{3}}} x} + \sin \theta} \tag{28}$$

The flow field distribution is as follows:

$$u_x = u_m e^{-\left( \frac{y^2}{\varepsilon^2 x^2} \right)} \tag{29}$$

$$u_y = \frac{\varepsilon^2 x e^{-\left( \frac{y^2}{\varepsilon^2 x^2} \right)}}{2y} \left[ u_m \left( 2 + x - \frac{x g n^2}{h^{\frac{4}{3}}} \right) + \frac{x g \sin \theta}{u_m} \right] \tag{30}$$

*2.3. Calculation of Erosion and Deposition of Bed Surface*

The deformation equation of the bed surface is established for the delta dominated by bedload:

$$C_m \frac{\partial z_s}{\partial t} + \frac{1}{\gamma_s} P_{bl} \left( \frac{\partial q_{blx}}{\partial x} + \frac{\partial q_{bly}}{\partial y} \right) = 0 \tag{31}$$

where $P_{bl}$ is the sediment composition of group $l$; $\gamma_s$ is specific gravity of sediment; $C_m$ is the maximum sediment concentration of muddy water to maintain fluid characteristics and can be calculated as follows [31]:

$$C_m \frac{\partial z_s}{\partial t} + \frac{1}{\gamma_s} P_{bl} \left( \frac{\partial q_{blx}}{\partial x} + \frac{\partial q_{bly}}{\partial y} \right) = 0 \tag{32}$$

where $d_l$ is in millimeters; $q_{blx} = q_{bl} \frac{u_x}{\sqrt{u_x^2 + u_y^2}}$, $q_{bly} = q_{bl} \frac{u_y}{\sqrt{u_x^2 + u_y^2}}$, and $q_{bl}$ is the bedload transport rate of group $l$, which can be calculated according to the method of Levy [31]:

$$q_{bl} = 2d_l \left( \frac{U}{\sqrt{g d_l}} \right)^3 (U - u_{lc}) \left( \frac{d_l}{h} \right)^{0.25} \tag{33}$$

where $U = \sqrt{u_x^2 + u_y^2}$, and $u_{lc}$ is the incipient velocity of bedload of group $l$ which can be calculated as follows [31]:

$$u_{lc} = (h/d_l)^{0.14} \sqrt{17.6(\gamma_s - \gamma)d_l/\gamma + 0.000000605(10 + h)/d_l^{0.72}} \tag{34}$$

where $\gamma$ is the specific gravity of water. The applicable range of Levy's formula is: $d_l = 0.25–23$ mm, $h/d_l = 5–500$, $U/u_{lc} = 1–3.5$.

The following can be obtained from the bed deformation equation:

$$\frac{\partial z_s}{\partial t} = \frac{-P_{bl}}{\gamma_s C_m} \left( \frac{\partial q_{blx}}{\partial x} + \frac{\partial q_{bly}}{\partial y} \right) = \frac{-P_{bl}}{\gamma_s C_m} \frac{2d_l}{\left( \sqrt{g d_l} \right)^3} \left( \frac{d_l}{h} \right)^{0.25} \left\{ \left[ 2\alpha \delta u_x + U^2 \left( \alpha \frac{u_x}{U} + \delta \frac{\partial u_x}{\partial x} \right) \right] + \left[ 2\beta \delta u_x + U^2 \left( \beta \frac{u_x}{U} + \delta \frac{\partial u_x}{\partial y} \right) \right] \right\} \tag{35}$$

where $\alpha = u_x \frac{\partial u_x}{\partial x} + u_y \frac{\partial u_y}{\partial x}$, $\beta = u_x \frac{\partial u_x}{\partial y} + u_y \frac{\partial u_y}{\partial y}$, $\delta = U - u_{lc}$.

The flow field distribution is taken into Equation (35) and accumulation rate can be obtained; that is, the change in erosion and deposition at the initial moment. Equation (35) reflects the variation characteristics of deposition thickness along the $x$-axis when hydraulic elements are constant. Since the above solution is the variation rate of the initial instantaneous bed deposition, which reflects the instantaneous erosion and deposition characteristics of the delta, no numerical iteration calculation has been carried out; that is, the impact of the erosion and deposition of the existing sediment on the next flow and sediment movement has not been taken into account, so it applies to the condition when the delta is fully developed and the impact of the initial deposition on the flow and sediment movement is still weak.

## 3. Validation of the Model

The experiment was carried out in Tianjin University. A flume (3.6 m long, 1.7 m wide, 0.2 m high), a water storage tank, experimental area, and circulation system were included in the experimental equipment, as shown in Figure 2.

The experimental lacustrine shallow-water delta was used for the verification of our theoretical model for the bedload dominance cases. The experiment was expected to reproduce a process similar to that occurring in reality during bedload movement in the formation and evolution of a delta. This required a low Froude number (e.g., the Froude number of Mississippi River is about 0.1; the Froude number of Gan River-Fu River trail delta is about 0.12; the Froude number of our delta is about 0.09) and was bedload-dominated, such as in terms of key trends in spatial and temporal patterns of flow and sediment movement and topography. The selection of inflow amount mainly considered that the upstream flow velocity should be larger than the sediment incipient velocity and the morphological evolution should be more remarkable in the predicted

duration. Therefore, the maximum value of the flowmeter of 100 cm$^3$/s was selected. The selection of grain size mainly considered that sediment particles were easy to flocculate if they were too fine, whereas it should not be too large combined with the selected discharge value; thus, a kind of natural sediment having a median grain size $d$ = 0.3 mm was selected in our experiment.

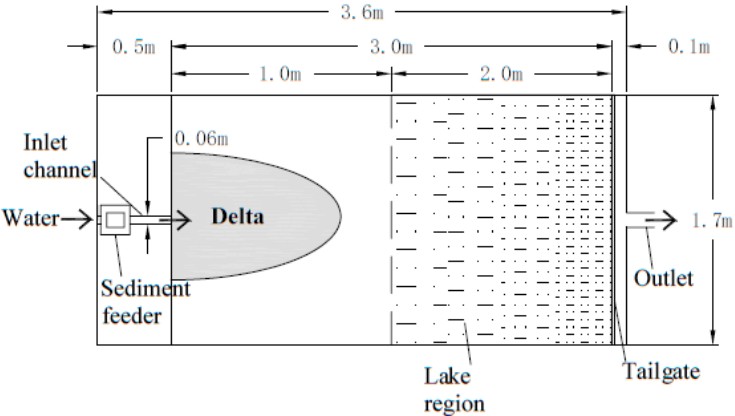

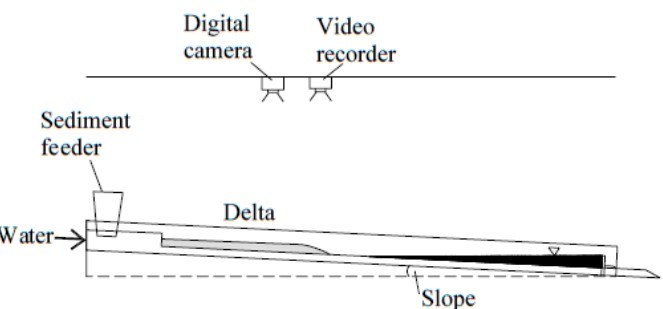

**Figure 2.** Experimental device diagram.

The sediment concentration is usually set at about 0.2% in shallow water delta experiments [32–34]. Considering that the amount of sediment coming into the upper reaches of a natural shallow-water delta is small, the sediment concentration was set at 0.2%; pre-experimental results showed that the increase in incision was not conducive to the formation of deposits when the slope was too large, so 1% was chosen as the slope. The tail gate can avoid the rapid outflow in the experimental area, and helps to make a lake area downstream of the flume to simulate the state of a river entering a lake. The tail gate height was set as 2 cm, which optimized the length of the flume.

The desired experimental slope can be obtained by adjusting the bottom height of the flume; this slope was set as 1%. A 5 cm thick sediment was laid at the bottom of the flume (non-cohesive and equal-diameter sediment with density $\rho_s$ = 2650 kg/m$^3$ and grain size $d$ = 0.3 mm was used at the entrance and the bottom of the flume) and can prevent the erosion from reaching the bottom.

After the start of the experiment, the sediment feeder above the artificial river began to make the sediment fall, which mixed with the flow in the river and moved to the outlet of the river and rushed out. The velocity of the flow rushing out of the estuary decreased and the sediment fell and deposited to make a delta. The sediment mainly moved as bedload under the designed discharge during the experimental process, like the delta dominated by bedload in nature. The model parameters are shown in Table 1.

**Table 1.** Parameter settings of the model.

| Quantity | Value |
| --- | --- |
| discharge Q (cm$^3$/s) | 100 |
| depth h (cm) | 2.0 |
| slope J (/) | 1% |
| particle size d (mm) | 0.3 |
| roughness n (/) | 0.02 |
| volumetric sediment supplied rate $S_v$ (/) | 0.2% |

The experiment was run until the end of the first hour, when the delta was fully developed and the data collected under the existing accuracy fully reflected the trend of topographic changes, and the impact of the initial deposit pattern on flow and sediment movement could be controlled as much as possible. We used a laser rangefinder to collect the terrain data, and the distance between the laser rangefinder and initial bed surface was kept constant to obtain terrain data.

It is not practicable to directly compare the experimental data with the calculation result of theoretical model because the experimental data reflect a time-average state of flow movement and sediment deposition, whereas the calculation results present the instantaneous state of flow and sediment. In addition, some parameters (such as roughness) that cannot be measured point-by-point are included in the theoretical model, and random factors at each point on the bed are inevitable (such as the conditions of the bed surface, incoming water, and sediment supplied at each point). Therefore, we focus on comparing the main trend of deposition predicted by the model with the trend reflected by the experimental data. The measured values of erosion and deposition at the end of 1 h were selected and converted to a dimensionless form. These values were compared with the calculation result of the model in which the delta grew fully. This also ensured that the data collected under the existing accuracy accurately reflected the trend of terrain change and the influence of the initial deposition on the flow and sediment movement could be reduced.

The deposit advanced about 0.5 m in the *x*-direction and about 0.2 m in the *y*-direction, with the *x*-axis as the symmetry axis at the end of 1 h. Therefore, the range of 0.5 m in the *x*-direction and 0.2 m in the *y*-direction (0.2 m on both sides of *x* = 0) was selected for verification. A verification section was set every 0.1 m along the *x*-axis direction and a verification point was set every 0.05 m in the *y*-axis direction. The layout of verification points is shown in Figure 3:

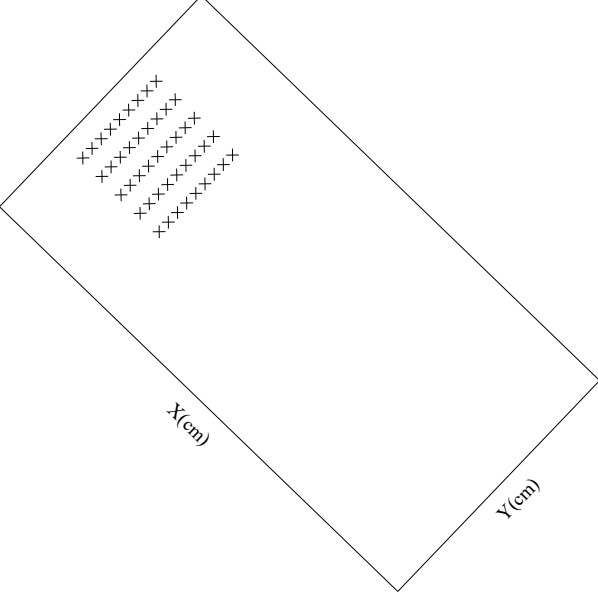

**Figure 3.** Schematic diagram of verification point layout.

In order to have a better understanding of the dynamic behavior of the system, it is worthwhile to rewrite $z_s/t$ and $z_s$ in dimensionless form. The following nondimensional variables are introduced:

$Zc^* = \frac{z_s/t}{S_v/(l\times b)}$, and $S_v/(l \times b)$ is "elevation scale of sediment supplied in unit time at artificial river", where $l$ is a length scale (e.g., characteristic length of artificial river) and $b$ is a width scale (e.g., characteristic width of artificial river), $Zm^* = \frac{z_s}{l}$, $Y^* = \frac{y}{l}$. Asterisk superscript represent dimensionless variables, the same below.

$Zc^*$ and $Zm^*$ of each section are shown in Figure 4. Through trial calculation, we found that the calculated values of each section were in best agreement with the measured values of the main deposition trend when $\varepsilon = 0.25$. It can be seen from this that, compared with clear water jet, the presence of sediment increases the value of $\varepsilon$, that is, the jet boundary of sediment-laden flow diffuses more widely along the straight line.

Based on the above experimental condition settings, the flow field distribution was calculated by the method proposed by this paper and that in Hydraulics of SCU [27]. The bed deformation was calculated by the same bed deformation equation, and the bed deformation results calculated by the two methods were compared. The axial velocity in Hydraulics of SCU [27] is for the main section ($x > 0.3$ m), so the two calculation methods were used to calculate the change rate of $z_s$ of sections $x = 0.4$ m and $x = 0.5$ m, as shown in Figure 4d–e.

In conclusion, our trend of the calculation result of the variation rate of $Zc^*$ is consistent with that of the measured value of $Zm^*$ at the end of 1 h from section $x = 0.1$ m to $x = 0.5$ m. The trends of calculated and measured values within the range of $x = 0.1$ m to $x = 0.2$ m are V-shaped, while that within the range of $x = 0.3$ m~$x = 0.5$ m are in a W-shape. The shape of the measured sectional topography develops more gently compared with that of the calculated values. The extension of the deepest scouring point appears on the $x = 0.3$ m section according to the measured value, and sectional asymmetric growth occurs on all sections.

It can be seen from Figure 4 that the trends of $Z^*$ of sections $x = 0.4$ m and $x = 0.5$ m by the two methods are generally consistent. The main difference is that the locations of the most severe scour points are different: the most severe scouring points appear at $y = \pm 0.1$ m in Hydraulics of SCU [27] and $y = \pm 0.05$ m in this paper. The experimental data show that the point with the largest scouring degree on section $x = 0.4$ m appears at $y = 0.05$ m, followed by $y = -0.05$ m; the minimum value of $z_s/t$ on the $x = 0.5$ m section lies at $y = -0.05$ m and $y = 0.2$ m, followed by $y = 0.05$ m. Combined with the analysis of experimental data, the calculation result in this paper is more reasonable.

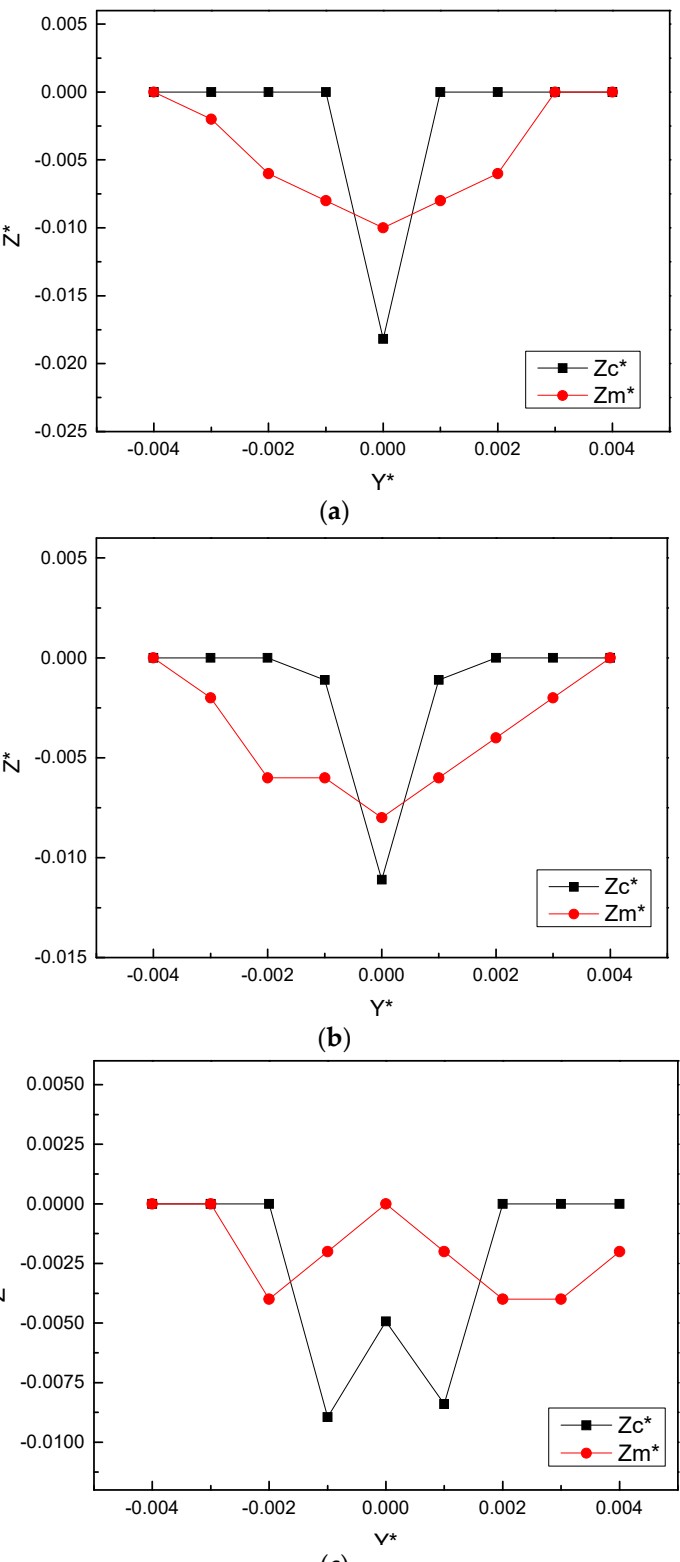

**Figure 4.** *Cont.*

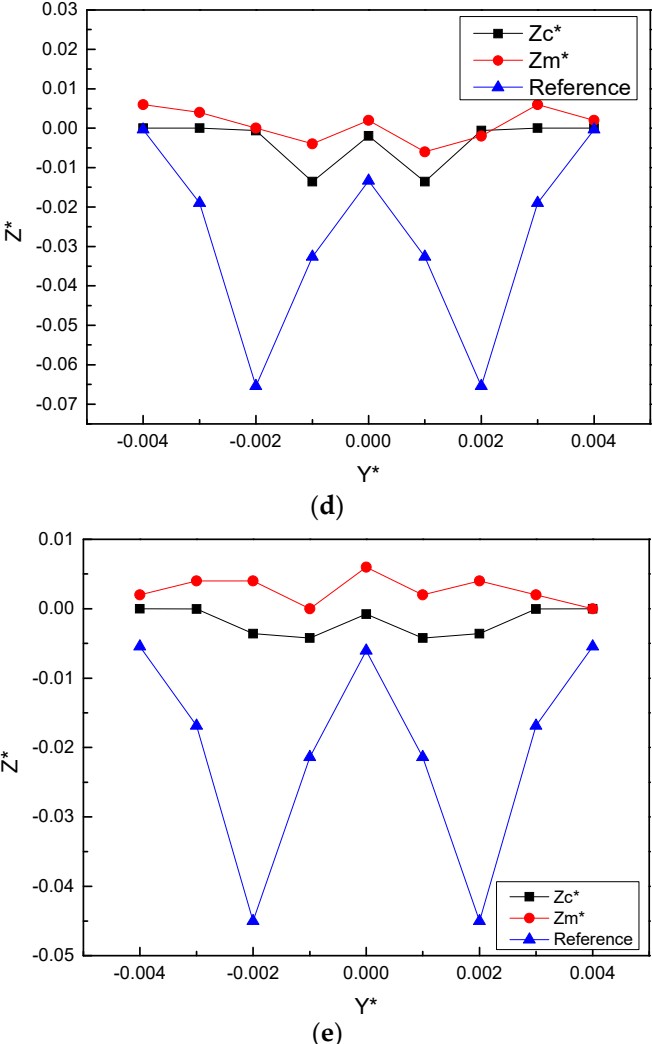

**Figure 4.** Calculated and measured values of erosion and deposition of each section. (**a**). Our solution $Zc^*$ and measured value $Zm^*$ of section of $x = 0.1$ m. (**b**). Our solution $Zc^*$ and measured value $Zm^*$ of section of $x = 0.2$ m. (**c**). Our solution $Zc^*$ and measured value $Zm^*$ of section of $x = 0.3$ m. (**d**). Our solution $Zc^*$ and measured value $Zm^*$ of section of $x = 0.4$ m. (**e**). Our solution $Zc^*$ and measured value $Zm^*$ of section of $x = 0.5$ m.

## 4. Conclusions

A theoretical model of the plane jet boundary layer in the initial stage of a delta is established based on the shallow-water equation of sediment-laden flow according to the characteristics of the movement of flow and sediment in the initial stage of delta growth. The similarity solution method is used and the flow field distribution is obtained. Our experimental data are used to calibrate the coefficient $\varepsilon$, which describes the characteristics of sediment-laden flow spreading along a straight line. The theoretical expression of the morphological characteristics of the initial stage of a lacustrine delta is derived. Experimental verification showed that the theoretical model can accurately reflect the rule of erosion and deposition of the delta in the initial stage.

It was found that the transitions between measured terrain points are gentler and smoother than the theoretical values, and the asymmetric growth appears in all sections. These findings are related to the greater kinetic energy possessed by the flow around the entrance and the act of flow turbulence on the non-uniform distribution of sediment (specific to a pinch of sediment) that amplifies the impact of the randomness of flow and sediment on the expression of delta morphology. Numerical hydrodynamic models are

not only a good complementary tool to test the suitability of a solution, but also a way to expand the application of the theoretical model to a wider range. This will be the next research direction for us.

**Author Contributions:** Conceptualization, Y.B. and Z.J.; methodology, W.X. (Weiyan Xin) and Y.Y.; validation, W.X. (Weiyan Xin), Y.Y. and W.X. (Wude Xie); writing—original draft preparation, W.X. (Weiyan Xin), Y.Y. and Z.J.; funding acquisition, Y.B. and Y.Y. All authors have read and agreed to the published version of the manuscript.

**Funding:** This research was funded by National Natural Science Foundation of China [Study on the service condition evaluation theory and function restoration intelligent decision method of the waterway regulation buildings in the middle and lower reaches of the Yangtze River], grant number [51979132].

**Conflicts of Interest:** The authors declare no conflict of interest.

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
