# Peer review of "Theoretical Model and Solution of Dynamic Evolution in Initial Stage of Lacustrine Delta"

_water, doi:10.3390/w14162522_

Round 1
Reviewer 1 Report
Please see the attached review document.

Author Response
Dear Reviewer:
Thank you for your comments concerning our manuscript entitled “Theoretical model and solution of dynamic evolution in initial stage of lacustrine delta” (Manuscript ID: Water-1724992). Those comments are all valuable and very helpful for revising and improving our paper, as well as the important guiding significance to our researches. We have studied comments carefully and have made correction which we hope meet with approval. Insert portions are marked in purple; deleted portions are marked in green with strikethrough. The main corrections in the paper and the responds to the reviewer’s comments are as flowing:
Reviewer#1:
- The authors have developed a theoretical model and its solution to dynamic evolution at the initial stage of lacustrine delta where the sediment “bed load” is dominant. The dynamic evolution at the initial stage of lacustrine delta has been studied by other researchers (e.g., Wang 1984 cited by the authors, Jiménez-Robles, et al. 2016*). The sediment transport when a river enters a lake or sea is commonly dominated by “suspended load” and this common case has been studied by other researchers (e.g., Wang 1984 cited by the authors). The suspended load and the bed load have also been considered together by other researchers (e.g., Jiménez-Robles, et al. 2016*). The authors’ study was for the case where the sediment “bed load” is dominant. It would be better for the authors to describe how common the case of bed-load dominance is. This would help convince the readers that their study is relevant and useful.
Response:
As suggested, in order to make the research necessity clearer, the sediment data and description of the Three Gorges Reservoir have been added and explained, which indicates that bedload is an important form of sediment movement into lake and it has an important impact on the formation and evolution of delta. According to the study of Chen etal (2013) and their estimation, the bedload and bottom sediment missed from June 2003 to December 2010 accounted for about 7.3% of the suspended sediment in the Three Gorges reservoir. Their study also shows that the proportion of bedload is affected by sediment particle size, water depth and flow velocity, that is, the proportion of bed load and suspended load changes with condition variation such as sediment particle size and water depth. Moreover, it is difficult to observe the bed load in practice, and it is easy to miss the measurement, resulting in the measured bed load sediment transport rate is small. Relatively speaking, it is undoubtedly basic and reliable to study the movement rule of bed load of the delta in the reservoir area from a theoretical point of view. See the revised manuscript (page1, line32)~(page2, line57).
- The hydrodynamic part alone and the sediment bed-load transport part alone of the authors’ model are not new although the combination of hydrodynamics and sediment “bed-load” transport for the authors’ specific study case is new. The hydrodynamic equations (Eq.16 for mass conservation, Eqs. 17 and 18 for momentum conservation) are the well-established steady, depth averaged, plane 2-D hydrodynamic equations and are readily available from the literature such as a textbook on computational hydraulics. The sediment transport equation (Eq. 35) is a typical stream bed degradation/aggregation equation that is also well established as indicated by the authors.
Response:
In order to study hydrodynamics characteristics and bedload movement rule, we combined hydrodynamics and bedload transport, and not only the analytical solution of the flow velocity in the direction of flow (x direction) was solved, but also that of the flow velocity in Y direction was given. We proposed ε to describe the characteristics of sediment laden flow jet spreading along a straight line, and its empirical value was also obtained, which has been added below Eqs (19). See the revised manuscript (page6, line197)~(page6, line205) and (page11, line356)~(page11, line360). And a lacustrine delta experiment were designed and carried out which is for the case where bedload is dominant to verify the theoretical model, and the theoretical model has been well verified.
- The model simplification and the subsequent similarity solution is not new either. Hydrodynamics of the river flow into a large, shallow water body was simplified to a plane jet problem by the authors. The similarity solution to the hydrodynamic equations was subsequently taken by the authors. This type of approach has been taken before (e.g., Wang 1984 cited by the authors, Jiménez-Robles, et al. 2016).
Response:
In the process of solving, we referred to the definition of empirical coefficient in characteristic half thickness calculation method of clear water jet theory and introduced ε to describe the characteristics of sediment laden flow jet spreading along a straight line, and its empirical value was also obtained through trial calculation. It can be seen from this that, the presence of sediment increases the value of ε, that is, the jet boundary of sediment laden flow diffuses more widely along the straight line than clear water jet. The above description has been added below Eqs.(19) and below Fig.4, see the revised manuscript (page6, line197)~(page6, line205) and (page11, line356)~(page11, line360).
- The authors’ laboratory test setup, procedures and data should have been described in more detail and more clearly. It should have been explained that the laboratory test condition was for the bed-load dominance case that the authors tried to model and thus was used to validate their “new” model.
Response:
Considerations on the determination of experimental conditions (flow, sediment particle size, sediment concentration, slope, tailgate height) have been added below Fig. 2. See the revised manuscript (page9, line277-line296).
After the start of the experiment, the sediment feeder above the artificial river began to make the sediment fall, and the sand mixed with the flow in the river and moved to the outlet of the river and rushed out. The velocity of flow rushing out of the estuary decreased and the sediment fell and deposited to make a delta. The above description of laboratory process has been added above Table 1. See the revised manuscript (page10, line308)~(page10, line311).
We used laser rangefinder to collect the terrain data, and the distance between laser rangefinder and initial bed surface was kept constant to get terrain data. The explanation of data has been added below Table 1. See the revised manuscript (page10, line323)~(page10, line325).
The experiment was expected to reproduce similar process that occurred during bedload movement in the formation and evolution of delta in reality, which requiring low Froude number and bedload dominated, such as key trends in spatial and temporal patterns of flow and sediment movement and topography, which has been added below Fig.2, see the revised manuscript (page9, line274)~(page9, line277).
- The authors need to elaborate on their statement (Line 221): “It [the model] applies to the condition when delta is fully developed while the impact of initial deposition on the flow and sediment movement is still weak.”
Response:
Since our solution is the variation rate of the initial instantaneous bed deposition, which not only reflects the instantaneous sedimentation characteristics, but also shows the scouring and silting trend in the initial stage to some extent. No numerical iteration calculation has been carried out, that is, the impact of the erosion and deposition of the existing sediment on the next flow and sediment movement has not been taken into account, so it applies to the condition when delta is fully developed while the impact of initial deposition on the flow and sediment movement is still weak. We added the explanation below Eqs.(40). See the revised manuscript (page8, line259)~(page8, line263)
- The authors need to explain why different dimensions/units of the lab-measured and the model-calculated quantities were used in the comparison (rate of change in bed elevation Zs/t vs. elevation Zs in Figure 5.). The rate of bed-elevation change (Zs/t) and the cumulative bed elevation (Zs) should both be compared but separately.
Response:
It is not practicable to compare the experimental data with the calculation result of theoretical model directly. That’s because the experimental data reflect the instantaneous state of flow and sediment while the calculation results present a time-average state of flow movement and sediment deposition.
However, the calculation results can describe the trend of erosion and deposition in initial stage, it is reasonable to compare the trend of Zs/t and measured terrain in the initial stage. In order to reveal the essential rule, the variation trend of the two (Zs/t vs Zs) in each section is compared after making them dimensionless and showed in Fig.5. Relevant explanations has been added below Fig.4, see the revised manuscript (page11, line349)~(page11, line354), and comparison results are shown in Fig.5.
Other changes:
- ‘The comparison of the calculation results between our model and Hydraulics of SCU (2016)’in original manuscript is combined with ‘the comparison between the calculated value and the measured value’.
- ‘The comparison of the delta axis topographic data between this paper and the research of Bai et al. (2018)’is deleted.
- We added some references:
(1) Chen, X., Zheng, B.M., Hu, C.H., 2013. Stochastic analysis of sediment movement in the Three Gorges Reservoir. Journal of Sediment Research 12, 6-11. See ( page1, line33)~(page2, line57).
(2) Clarke, L.E., Quine, T.A., Nicholas, A.P., 2010. An experimental investigation of autogenic behaviour during alluvial fan evolution. Geomorphology 115, 278–285. See ( page9, line289).
(3) Hoyal, D.C.J.D., Sheets, B.A., 2009. Morphodynamic evolution of experimental cohesive deltas. J. Geophys. Res. Earth Surf. 114 (F2). See ( page9, line289).
(4) Jimenez-Robles, A. M , Ortega-Sanchez, et al., 2016. Effects of basin bottom slope on jet hydrodynamics and river mouth bar formation. Journal of geophysical research: Earth Surface 121, 1110-1133. See ( page2, line95-line97).
(5) Zhang, X.F., Wang, S.Q., Wu, X., Xu, S., Li, Z.Y., 2016. The development of a laterally confined laboratory fan delta under sediment supply reduction. Geomorphology 257, 120–133. See ( page9, line289).
Reference
Chen, X., Zheng, B.M., Hu, C.H., 2013. Stochastic analysis of sediment movement in the Three Gorges Reservoir. Journal of Sediment Research 12, 6-11.
We tried our best to improve the manuscript and made some changes in the manuscript. These changes will not influence the content and framework of the paper. And the changes are marked in the revised manuscript. We appreciate for Editors/Reviewers’ warm work earnestly, and hope that the correction will meet with approval.
Once again, thank you very much for your comments and suggestions.

Reviewer 2 Report
I am not satisfied with the validation of the model section. Computed and measured data must be shown in a same graph. In that case, you would have five graphs for x = 0.1 m, 0.2 m, 0.3 m, 0.4 m, 0.5 m. Regarding the magnitude issue, my suggestion is to convert the data in non-dimensional form and then present it in a graph. It can give a clear picture of comparison and the validation of the model.
Author Response
Dear Reviewer:
Thank you for your comments concerning our manuscript entitled “Theoretical model and solution of dynamic evolution in initial stage of lacustrine delta” (Manuscript ID: Water-1724992). Those comments are all valuable and very helpful for revising and improving our paper, as well as the important guiding significance to our researches. We have studied comments carefully and have made correction which we hope meet with approval. Insert portions are marked in purple; deleted portions are marked in green with strikethrough. The main corrections in the paper and the responds to the reviewer’s comments are as flowing:
Reviewer#2:
- I am not satisfied with the validation of the model section. Computed and measured data must be shown in a same graph. In that case, you would have five graphs for x = 0.1 m, 0.2 m, 0.3 m, 0.4 m, 0.5 m. Regarding the magnitude issue, my suggestion is to convert the data in non-dimensional form and then present it in a graph. It can give a clear picture of comparison and the validation of the model.
Response:
As suggested, we converted the data (including calculated result of Zs/t with our model and method mentioned in the reference, and measured value of Zs in our experiment) in non-dimensional form and presented it in a graph. Relevant explanations has been added below Fig.4, see the revised manuscript (page11, line349)~(page11, line354), and comparison results are shown in Fig.5.
Other changes:
- In order to make the research necessity clearer, the sediment data and description of the Three Gorges Reservoir have been added and explained, which indicates that bedload is an important form of sediment movement into lake and it has an important impact on the formation and evolution of delta. According to the study of Chen etal (2013)and their estimation, the bedload and bottom sediment missed from June 2003 to December 2010 accounted for about 7.3% of the suspended sediment in the Three Gorges reservoir. Their study also shows that the proportion of bedload is affected by sediment particle size, water depth and flow velocity, that is, the proportion of bed load and suspended load changes with condition variation such as sediment particle size and water depth. Moreover, it is difficult to observe the bed load in practice, and it is easy to miss the measurement, resulting in the measured bed load sediment transport rate is small. Relatively speaking, it is undoubtedly basic and reliable to study the movement rule of bed load of the delta in the reservoir area from a theoretical point of view. See the revised manuscript (page1, line32)~(page2, line57).
- In the process of solving, we referred to the definition of empirical coefficient in characteristic half thickness calculation method of clear water jet theory and introduced εto describe the characteristics of sediment laden flow jet spreading along a straight line, and its empirical value was also obtained through trial calculation. It can be seen from this that, the presence of sediment increases the value of ε, that is, the jet boundary of sediment laden flow diffuses more widely along the straight line than clear water jet. The above description has been added below Eqs.(19) and below Fig.4, see the revised manuscript (page6, line197-line205) and (page11, line356-line360).
- Considerations on the determinationof experimental conditions (flow, sediment particle size, sediment concentration, slope, tailgate height) have been added belowFig. 2. See the revised manuscript (page9, line277-line296).
After the start of the experiment, the sediment feeder above the artificial river began to make the sediment fall, and the sand mixed with the flow in the river and moved to the outlet of the river and rushed out. The velocity of flow rushing out of the estuary decreased and the sediment fell and deposited to make a delta. The above description of laboratory process has been added above Table 1. See the revised manuscript (page10, line308)~(page10, line311).
We used laser rangefinder to collect the terrain data, and the distance between laser rangefinder and initial bed surface was kept constant to get terrain data. The explanation of data has been added below Table 1. See the revised manuscript (page10, line323)~(page10, line325).
The experiment was expected to reproduce similar process that occurred during bedload movement in the formation and evolution of delta in reality, which requiring low Froude number and bedload dominated, such as key trends in spatial and temporal patterns of flow and sediment movement and topography, which has been added below Fig.2, see the revised manuscript (page9, line274)~(page9, line277).
- 5.Since our solution isthe variation rate of the initial instantaneous bed deposition, which not only reflects the instantaneous sedimentation characteristics, but also shows the scouring and silting trend in the initial stage to some extent. No numerical iteration calculation has been carried out, that is, the impact of the erosion and deposition of the existing sediment on the next flow and sediment movement has not been taken into account, so it applies to the condition when delta is fully developed while the impact of initial deposition on the flow and sediment movement is still weak. We added the explanation below Eqs.(40). See the revised manuscript (page8, line259)~(page8, line263)
- ‘The comparison of the calculation results between our model and Hydraulics of SCU (2016)’in original manuscript was combined with ‘the comparison between the calculated value and the measured value’.
- ‘The comparison of the delta axis topographic data between this paper and the research of Bai et al. (2018)’was deleted.
- We added some references:
(1) Chen, X., Zheng, B.M., Hu, C.H., 2013. Stochastic analysis of sediment movement in the Three Gorges Reservoir. Journal of Sediment Research 12, 6-11. See ( page1, line33)~(page2, line57).
(2) Clarke, L.E., Quine, T.A., Nicholas, A.P., 2010. An experimental investigation of autogenic behaviour during alluvial fan evolution. Geomorphology 115, 278–285. See ( page9, line289).
(3) Hoyal, D.C.J.D., Sheets, B.A., 2009. Morphodynamic evolution of experimental cohesive deltas. J. Geophys. Res. Earth Surf. 114 (F2). See ( page9, line289).
(4) Jimenez-Robles, A. M , Ortega-Sanchez, et al., 2016. Effects of basin bottom slope on jet hydrodynamics and river mouth bar formation. Journal of geophysical research: Earth Surface 121, 1110-1133. See ( page2, line95-line97).
(5) Zhang, X.F., Wang, S.Q., Wu, X., Xu, S., Li, Z.Y., 2016. The development of a laterally confined laboratory fan delta under sediment supply reduction. Geomorphology 257, 120–133. See ( page9, line289).
Reference
Chen, X., Zheng, B.M., Hu, C.H., 2013. Stochastic analysis of sediment movement in the Three Gorges Reservoir. Journal of Sediment Research 12, 6-11.
We’ve tried our best to improve the manuscript and made some changes in the manuscript. These changes will not influence the content and framework of the paper. And the changes are marked in the revised manuscript. We appreciate for Editors/Reviewers’ warm work earnestly, and hope that the correction will meet with approval.
Once again, thank you very much for your comments and suggestions.

Round 2
Reviewer 1 Report
The authors have addressed all my comments.
Reviewer 2 Report
The comparison between non-dimensional values looks good. I recommend to accept your paper for publication.